# Class III Peroxidases PRX01, PRX44, and PRX73 Control Root Hair Growth in *Arabidopsis thaliana*

**DOI:** 10.3390/ijms23105375

**Published:** 2022-05-11

**Authors:** Eliana Marzol, Cecilia Borassi, Mariana Carignani Sardoy, Philippe Ranocha, Ariel A. Aptekmann, Mauro Bringas, Janice Pennington, Julio Paez-Valencia, Javier Martínez Pacheco, Diana R. Rodríguez-Garcia, Yossmayer del Carmen Rondón Guerrero, Juan Manuel Peralta, Margaret Fleming, John W. Mishler-Elmore, Silvina Mangano, Francisca Blanco-Herrera, Patricia A. Bedinger, Christophe Dunand, Luciana Capece, Alejandro D. Nadra, Michael Held, Marisa S. Otegui, José M. Estevez

**Affiliations:** 1Fundación Instituto Leloir and IIBBA-CONICET. Av. Patricias Argentinas 435, Buenos Aires C1405BWE, Argentina; emarzol@leloir.org.ar (E.M.); ceciborassi@gmail.com (C.B.); mcarignani@leloir.org.ar (M.C.S.); jmartinez@leloir.org.ar (J.M.P.); drodriguez@leloir.org.ar (D.R.R.-G.); yrondon@leloir.org (Y.d.C.R.G.); jperalta@leloir.org.ar (J.M.P.); silvinamangano@gmail.com (S.M.); 2Laboratoire de Recherche en Sciences Végétales, Université de Toulouse, CNRS, UPS, Toulouse INP, 24, Chemin de Borde-Rouge, 31320 Auzeville-Tolosane, France; ranocha@lrsv.ups-tlse.fr (P.R.); dunand@lrsv.ups-tlse.fr (C.D.); 3Departamento de Fisiología, Biología Molecular y Celular, Instituto de Biociencias, Biotecnología y Biología Traslacional (iB3). Facultad de Ciencias Exactas y Naturales, Universidad de Buenos Aires, Ciudad Universitaria, Buenos Aires C1428EGA, Argentina; arielaptekmann@gmail.com (A.A.A.); anadra@qi.fcen.uba.ar (A.D.N.); 4Departamento de Química Biológica, Facultad de Ciencias Exactas y Naturales, Universidad de Buenos Aires (IQUIBICEN-CONICET), Ciudad Universitaria, Buenos Aires C1428EGA, Argentina; 5Departamento de Química Inorgánica, Analítica y Química Física, Facultad de Ciencias Exactas y Naturales, Universidad de Buenos Aires (INQUIMAE-CONICET), Buenos Aires C1428EGA, Argentina; maubringas@gmail.com (M.B.); lula.capece@gmail.com (L.C.); 6Laboratory of Cell and Molecular Biology, University of Wisconsin, Madison and Center for Quantitative Cell Imaging, University of Wisconsin, Madison, WI 53706, USA; pennington2@wisc.edu (J.P.); paezvalencia@wisc.edu (J.P.-V.); otegui@wisc.edu (M.S.O.); 7Department of Biology, Colorado State University, Fort Collins, CO 80523-1878, USA; mbfleming@gmail.com (M.F.); patricia.bedinguer@colostate.edu (P.A.B.); 8Department of Chemistry and Biochemistry, Ohio University, Athens, OH 45701, USA; je149004@ohio.edu (J.W.M.-E.); held@ohio.edu (M.H.); 9Center of Applied Ecology and Sustainability (CAPES), Santiago 8320000, Chile; mblanco@unab.cl; 10Departments of Botany and Genetics, University of Wisconsin, Madison, WI 53706, USA; 11Centro de Biotecnología Vegetal, Facultad de Ciencias de la Vida, Universidad Andrés Bello Santiago, Santiago 8370146, Chile; 12ANID—Millennium Science Initiative Program—Millennium Institute for Integrative Biology (iBio) and Millennium Nucleus for the Development of Super Adaptable Plants (MN-SAP), Santiago 8370146, Chile

**Keywords:** *Arabidopsis*, class-III peroxidases, cell walls, extensins, root hairs, ROS

## Abstract

Root hair cells are important sensors of soil conditions. They grow towards and absorb water-soluble nutrients. This fast and oscillatory growth is mediated by continuous remodeling of the cell wall. Root hair cell walls contain polysaccharides and hydroxyproline-rich glycoproteins, including extensins (EXTs). Class-III peroxidases (PRXs) are secreted into the apoplastic space and are thought to trigger either cell wall loosening or polymerization of cell wall components, such as Tyr-mediated assembly of EXT networks (EXT-PRXs). The precise role of these EXT-PRXs is unknown. Using genetic, biochemical, and modeling approaches, we identified and characterized three root-hair-specific putative EXT-PRXs, PRX01, PRX44, and PRX73. *prx01,44,73* triple mutation and PRX44 and PRX73 overexpression had opposite effects on root hair growth, peroxidase activity, and ROS production, with a clear impact on cell wall thickness. We use an EXT fluorescent reporter with contrasting levels of cell wall insolubilization in *prx01,44,73* and PRX44-overexpressing background plants. In this study, we propose that PRX01, PRX44, and PRX73 control EXT-mediated cell wall properties during polar expansion of root hair cells.

## 1. Introduction

Primary cell walls are composed by polysaccharides and glycoproteins. Extensins (EXTs) belong to hydroxyproline (Hyp)-rich glycoprotein superfamily (HRGP) with multiple Ser-(Pro)_3–5_ repeats to be *O*-glycosylated and Tyr (Y)-based motifs [1,2]. EXTs require several post-transductional modifications before becoming functional [1,2]. After being hydroxylated and *O*-glycosylated in the secretory pathway, they are secreted and insolubilized in the cell wall. EXTs are crosslinked by class-III peroxidases (PRXs) in the Tyr-based motifs [1,2,3,4,5,6]. PRXs are thought to facilitate both intra- and inter-molecular covalent Tyr–Tyr crosslinks in EXT networks, possibly through the assembly of triple helices [2,7] by generating isodityrosine units (IDT) and pulcherosine or di-isodityrosine (Di-IDT), respectively [6,8,9]. In addition, the O-glycosylation levels in EXTs affect the insolubilization process in the cell wall [7], likely by modulating EXT interactions with other cell wall components [10,11]. However, the underlying molecular mechanisms of EXT crosslinking and assembly have not been fully determined. It is proposed that *O*-glycosylation levels, as well as the presence of Tyr-mediated crosslinking in EXT and related glycoproteins, would allow them to form a dendritic glycoprotein network in the cell wall. This EXT network modulates de novo cell wall formation during embryo development [12,13] and regulates roots, petioles, and rosette leaves growth [14,15] and polar cell expansion of root hairs [3,4,5,7,16,17,18] and pollen tubes [19,20,21].

Class-III PRXs are heme-iron-dependent proteins, mainly secreted and members of a large multigenic family in land plants, with 73 members in *Arabidopsis thaliana* [22,23]. These PRXs catalyze several different classes of reactions. PRX activities coupled to _apo_ROS molecules (_apo_H_2_O_2_) directly affect the degree of cell wall crosslinking [24] by oxidizing cell wall compounds and leading to stiffening of the cell wall through a peroxidative cycle (PC) [1,22,25]. In contrast, _apo_ROS coupled to PRX activity enhances nonenzymatic cell wall loosening by producing oxygen radical species (e.g., ^●^OH) and promoting growth in the hydroxylic cycle (HC). In this HC cycle, PRXs catalyze the release of hydroxyl radicals (^●^OH) from H_2_O_2_ after O2^●-^ dismutation. For example, PRX36 has been reported to weaken plant cell walls in seed mucilage secreted by Arabidopsis seed coat epidermal cells [26]. It is unclear how these two opposite effects on cell wall polymers are co-ordinated during plant growth [22,25,27,28,29]. Finally, PRXs also contribute to the superoxide radical (O2^●−^) pool by oxidizing singlet oxygen in the oxidative cycle (OC), thereby affecting _apo_H_2_O_2_ levels. Thus, several PRXs are involved in the oxidative polymerization of monolignols in the apoplast of the lignifying xylem cells (e.g., PRX17 [30] and PRX72 [31]) and in the root endodermis (e.g., PRX64; [27,28]). In addition, PRXs are able to polymerize other components of the plant cell wall, such as suberin [32], pectins [29], and EXTs [33,34]. Although several candidates of PRXs have been implicated specifically in EXT crosslinking (EXT-PRXs) by in vitro studies [33,34,35,36,37,38] or by extensin immunolabelling [39], the in vivo characterization and mode of action of these EXT-PRXs remain largely unknown. Recently, PRX62 and PRX69 were identified as key apoplastic PRXs that modulate ROS homeostasis and cell wall EXT insolubilization linked to root hair elongation at low temperature [40]. In addition, PRX44 was implicated in ROS accumulation linked to the root hair initiation process [41]. In addition, fine-tuned cell wall modifications during cell expansion mediated by ethylene and auxin signaling pathways might involve class-III PRXs linked to ROS homeostasis [42,43,44]. In this work, by using a combination of reverse genetics, molecular and cell biology, computational molecular modeling, and biochemistry, we identified three class-III apoplastic PRXs, PRX01, PRX44, and PRX73, as key enzymes likely involved in the insolubilization of cell wall EXTs in growing root hair cells. Our results bring new insights in the organization of EXT assemblies during root hair development and provides a reference framework on how similar EXT networks may assemble in other cell types or in response to the environmental changes, such as fluctuating nutrient availability in the soil.

## 2. Results

### 2.1. Class-III PRX01, PRX44, and PRX73 Are Required for ROS-Mediated Root Hair Growth

Root hair cells are an excellent model for tracking cell elongation and identifying PRXs involved in cell wall EXT-mediated assembly. In previous work, the analysis of single mutants for PRX01, PRX44, and PRX73 suggested that these are involved in root hair growth and ROS homeostasis, although their mechanisms of action remained to be characterized [42]. These three PRXs are regulated by transcription factor RSL4 [42,44] and are highly co-expressed with other root-hair-specific genes, such as cell wall EXTs (e.g., EXT6-7, EXT12-14, and EXT18) and EXT-related glycoproteins (e.g., LRX1 and LRX2) involved in cell expansion [5,16,18] (Appendix A). Based on this co-expression evidence, we hypothesized that these three PRXs might be implicated on the assembly of EXTs in root hair cell walls. We then validated that PRX01, PRX44, and PRX73 are specifically expressed in root hairs using transcriptional reporters (Figure 1a). A previous report suggested functional redundancy between PRX01, PRX44, and PRX73 in RH growth because single mutants of these three PRXs showed normal root hair [42]. So, in this work, we analyzed the triple null mutant, *prx01 prx44 prx73* (*prx01,44,73*), that showed shorter root hairs than Wt Col-0 (Figure 1b) and 50% reduced root peroxidase activity in vitro (see Materials and Methods) (Figure 1c). Both results are consistent with short RH phenotype and reduced root peroxidase activity in seedlings treated with salicylic hydroxylamine acid (SHAM), a total peroxidase activity inhibitor [42]. We hypothesized that these three PRXs might change the levels of ROS, most probably H_2_O_2_, through their catalytic functions in the cell wall/apoplast. The homeostasis and levels of ROS (mostly H_2_O_2_) that regulate polar growth of root hair cells [42] involve apoplastic ROS _(apo_ROS) and cytoplasmatic ROS pools (_cyt_ROS). Dynamic exchange between the two ROS pools is mediated by their transport across the plasma membrane via specific aquaporins (PIPs for plasma membrane intrinsic proteins) in plant cells [45,46,47]. We measured cytROS levels by oxidation of H2DCF-DA and apoROS levels with the Amplex Ultra Red (AUR) probe in root hair tips. The *prx01,44,73* root hair tips showed lower levels of _cyt_ROS (Figure 1d) but increased apoROS accumulation (Figure 1e) compared to Wt Col-0.

Then, we analyzed the effect of overexpression of PRXs fused to GFP under the control of a constitutive 35SCaMV promoter (35S:PRX-GFP) (Figure 2a–c). 35S:PRX44-GFP and 35S:PRX73-GFP lines had significantly longer root hairs than the Wt Col-0 control. 35S:PRX01-GFP lines, however, had similar RH growth to Wt Col-0 (Figure 2a,b). In addition, 35S:PRX73-GFP roots showed a 45% increase in the in vitro peroxidase activity, 35S:PRX44-GFP presented 20% increase, whereas 35S:PRX01-GFP showed Wt col-0 levels of peroxidase activity (Figure 2e). The lack of root hair growth enhancement by 35S:PRX01-GFP might be due to regulatory aspects depending on the protein activity rather than on its abundance. Then, we tested if the overexpression of a single peroxidase is able to rescue root hair defects in the *prx01,44,73* triple mutant. Only 35S:PRX44-GFP, but not 35S:PRX01-GFP or 35S:PRX73-GFP, was able to restore normal root hair growth (Figure 2d). The _apo_ROS levels were similar in 35S:PRX01-GFP and slightly lower in 35S:PRX44, and 35S:PRX73-GFP root hairs when compared to Wt Col-0 (Figure 2f). These results suggest that PRX01, PRX44, and PRX73 function as apoplastic regulators of ROS-linked root hair cell elongation. This highlights that PRX01, PRX44, and PRX73 are partially redundant in their role as positive regulators of polar root hair growth.

### 2.2. Changes in prx01,44,73 Affect Cell Wall Structure and EXT Retention in Root Hairs

To analyze the impact of the peroxidase activity derived from PRX01, PRX44, and PRX73 on the ultrastructure of the cell wall, we take microphotographs of Wt and *prx01,44,73* triple mutant root hairs by transmission electron microscopy (Figure 3a). We found thinner cell walls in root hair tips of *prx01,44,73* (0.61 SD 0.14 μm) compared to Wt controls (1.2 SD 0.3 μm Wt) (Figure 3b). This result highlights the importance of peroxidase activity in cell wall structure. The depletion of PRX01, PRX44, and PRX73 (in the triple mutant *prx01,44,73*) results in the reduction in cell wall thickness in growing root hairs.

This implies that mis-regulation of PRX activity affects the capacity of root hairs to form normal cell walls and, as a consequence, affecting their cell elongation (Figure 2b).

Then, we designed an EXT reporter to track EXT secretion and PRX-mediated EXT retention in the cell walls during root hair cell elongation. This new reporter consists of a signal peptide, pdTomato, linked to EXT domain containing Tyr-based motifs (SS-TOM-EXT-LONG). We used TOM because it is fluorescent even under acidic pH [48] typical of plant cell walls and apoplastic spaces [49]. A similar construct lacking the EXT domain (SS-TOM) was used as a control (Appendix A). To verify the functionality of these reporters, they were expressed transiently in onion (*Allium cepa*) cells roots, and then stably expressed in Arabidopsis root hairs (Appendix A). In both cases, plasmolysis was used to verify that EXT reporter had been secreted and localized in cell walls. By immunoblot analysis, we observed that SS-TOM-EXT-LONG fusion protein had higher molecular weights than expected amino acid chain weight (Appendix A), consistent with the presence *O*-glycan modifications. Importantly, the EXT reporter did not interfere with polar growth of RH (Appendix A), making it an ideal probe to track in situ changes in the arrangement of cell wall EXTs. So, we tested if EXT reporter is differentially affected in *prx01,44,73* triple mutant and in 35S:PRX44 background. In both cases, we were detecting different levels of SS-TOM-EXT-LONG cell wall retention. It was higher in *prx01,44,73* triple mutant and slightly lower in the 35S:PRX44 background (Figure 4a,b). This suggests that changes in the peroxidase activity due to the suppression of several PRXs or the enhancement in expression of PRX44 modified the cell wall insolubilization of SS-TOM-EXT-LONG reporter. This is in agreement with the drastic changes in cell wall thickness in the *prx01,44,73* triple mutant studied by TEM (Figure 3). These results link changes in the ultrastructure of the walls with the retention of SS-TOM-EXT-LONG reporter, both derived from abnormal cell wall peroxidase activity levels.

We then assessed the level of crosslinking of EXT Tyr residues by measuring peptidyl-tyrosine (Tyr) and isodityrosine (IDT, dimerized Tyr) in EXT extracted from whole roots. We detected a significant increase in peptidyl-Tyr in the *prx01,44,73* triple mutant relative to Wt, and slightly higher levels of IDT in EXTs extracted from the PRX73-OE line (Table 1). By contrast, we identified a strong reduction in Tyr and IDT levels in the cell wall extracts from the *p4h5 sergt1-1* and *sergt1-1 rra3* mutants with reduced *O*-glycosylation in EXTs and related cell wall glycoproteins (Table 1). In these two double mutants, root hair growth is drastically inhibited [7]. These results are consistent with the notion that *O*-glycans strongly affect EXT Tyr crosslinking, as was previously suggested based on the drastically reduced root hair growth of the under-glycosylation mutants and in vitro crosslinking rates [7,18,50].

We hypothesize that absent or low *O*-glycosylation of EXTs or an increase in PRX levels may trigger a reduction in the amount of peptidyl-Tyr and IDT levels in EXTs, with a putative concomitant increase in the amounts of higher-order Tyr crosslinks (trimers as Pulcherosine and tetramers as Di-IDT), thus inhibiting root hair growth. Unfortunately, we were unable to measure the Pulcherosine and Di-IDT levels described previously in EXTs (Brady et al., 1996; 1998; Held et al., 2004) to test this hypothesis. Further research is needed to decipher the in vivo regulation of Tyr crosslinking of EXTs by these three PRXs in plant cells.

A major limitation in our understanding of how EXTs function and how they are modified by PRXs in plant cell walls is the lack of a realistic full-length EXT protein model. We used coarse-grained molecular dynamics to build a larger model of a triple-helix EXT sequence that includes 10 conserved repeats (SPPPPYVYSSPPPPYYSPSPKVYYK, 250 amino acids in each polypeptide chain) (Appendix A). Parameters for the *O*-glycosylated form of EXT were developed in this work (Appendix A). The EXT molecules were modeled in two different states: as a non-*O*-glycosylated trimeric helical conformation similar to animal collagen and in the *O*-glycosylated state, with four arabinose monosaccharide units in each hydroxyproline. The results indicate the importance of the triple-helix conformation in the overall stability of the protein, and especially in the conservation of its fibril-like structure, in agreement with shorter-repeats single-helix simulations performed previously [2,7]. The total volume of the extended system’s triple helix was measured in both glycosylation states (Appendix A), differentiating EXT-protein-only and EXT-protein + glycan volumes for the fully *O*-glycosylated EXT state. We observed that the EXT-protein-only volume was significantly augmented by the presence of the oligosaccharide moieties, indicating that *O*-glycans increase the distance between peptide chains in the EXT triple helix. We report the average diameters for those systems (Appendix A), which are consistent with those previously reported [51]. Additionally, *O*-glycosylation contributes to an increase in the average distance between the side chains of Tyr residues, decreasing the proportion of Tyr side chains that are close enough to lead to crosslinked EXT chains (Appendix A). Both experimental and modeling-based evidence are in agreement with the proposed role of proline-hydroxylation and carbohydrate moieties in keeping the EXT molecule in an extended helical polyproline-II conformation state [52,53,54]. This extended conformation might promote EXTs to interact with each other and with other components in the apoplast, including PRXs, to form a proper cell wall network [10,11].

### 2.3. PRX01, PRX44, and PRX73 Might Interact with EXTs

To test if these three PRXs (PRX01, PRX44, and PRX73) might be able to interact with single-chain EXTs, we performed homology modeling with GvEP1, an EXT-PRX that is able to crosslink EXTs in in vitro conditions [34,43], and two Arabidopsis apoplastic EXT-PRXs, PRX62 and PRX69, which modulate ROS homeostasis and cell wall EXT insolubilization linked to root hair elongation specifically at low temperature [40]. As controls, we included PRX64, a PRX implicated in lignin polymerization at the root endodermis [27], and PRX36, which binds homogalacturonan pectins in the seed coat [29]. By docking analysis, we obtained interaction energies (Kcal/mol) for all of them. We analyzed docking with four different short EXT peptides: a non-hydroxylated peptide, a hydroxylated peptide, an arabinosylated peptide, and an arabino-galactosylated peptide. It was previously shown that mutants carrying under-*O*-glycosylated EXTs have severe defects in root hair growth [7,16]. Our docking results for the different PRXs show consistent interaction energy differences that depend on the EXT glycosylation state, being higher for non-*O*-glycosylated species. In addition, *O*-glycosylated EXT variants docked in a rather dispersed way, while non-*O*-glycosylated variants preferentially docked in a grooved area (Appendix A). Furthermore, Appendix A shows how a non-*O*-glycosylated peptide binds through a groove, leaving one Tyr docked in a cavity and very close to the heme iron (5Å), with a second Tyr a few Angstroms away. The arrangement and distances between the Tyr residues suggest that this could be an active site where Tyr crosslinking takes place. Although it is not possible to compare the interaction energies obtained with the different EXT species among docking runs, a general trend can be observed in Appendix A. In general, we observed higher interaction energies (more negative values) for hydroxylated EXT species and lower for *O*-glycosylated EXT variants. When we compared interaction energies among different PRXs interacting with EXT substrates with the same degree of *O*-glycosylation, we observed that PRX73 displayed the highest interaction activity with the non-hydroxylated EXT species. PRX44 displayed the highest interaction energy with the *O*-glycosylated species. Altogether, these results are consistent with the constitutive root hair growth effect observed for 35S:PRX44 and 35S:PRX73 and with non-glycosylated EXT being the substrate of peroxidation. Overall, this possibly indicates that PRX44 and PRX73 might interact with EXT substrates and possibly catalyze Tyr crosslinking in open regions of the EXT backbones with little or no *O*-glycosylation. This is in agreement with previous studies that suggested that high levels of *O*-glycosylation in certain EXT segments physically restrict EXT lateral alignments, possibly by acting as a branching point [2,7,51].

### 2.4. Phylogenetic Insights of PRX01, PRX44, and PRX73

To examine if these putative EXT-PRXs (PRX01, PRX44, and PRX73) may have evolved together as a protein cluster, we performed comprehensive phylogenetic analyses of class-III peroxidases across diverse land plant lineages. Under low selective pressure, to maintain substrate specificity, EXT-PRX activities might have evolved multiple times during land plant evolution through gene duplication followed by neofunctionalization or subfunctionalization. PRX01, PRX44, and PRX73 belong to three independent orthologous groups (Appendix A), and orthologs for each *A. thaliana* PRX have been identified in available Brassicaceae genomes and in various Angiosperm and Gymnosperm families, but not in Lycophytes and Bryophytes. Thus, these three PRX sequences were the result of ancestral duplications before the divergence of Gymnosperms and Angiosperms but after the emergence of the Tracheophytes (Appendix A). Orthologs of the three PRX genes have only been detected in organisms with true roots (except lycophytes) and these three PRXs are expressed in roots and root hairs, as are most of their orthologous sequences (where expression data are available) (Appendix A). This strongly supports the hypothesis that the three independent orthogroups have conserved functions in roots. With the exception of PRX73, which belongs to a cluster containing a putative EXT-PRX from tomato (*Solanum lycopersicum*; LePRX38), the other two Arabidopsis PRX sequences did not cluster with any other predicted EXT-PRXs, such as PRX09 and PRX40 (Jacobowitz et al., 2019). Indeed, the other known EXT-PRXs (identified mostly based on in vitro evidence) did not cluster together, but were widely distributed in the tree (Appendix A). This analysis suggests that plant EXT-PRXs might have evolved several times during Tracheophyte evolution.

## 3. Discussion

### 3.1. Proposal of how PRX01, PRX44, and PRX73 Functions in the Root Hair Cell Walls

Based on the results shown in this work, we propose a working model in which PRX01, PRX44, and PRX73 (and possibly other PRXs) control root hair growth at room temperature by channeling H_2_O_2_ consumption and affecting the cell wall hardening process. In these polar growing cells, it is known that H_2_O_2_ is primarily derived from the respiratory burst oxidase homolog C (RBOHC) and, to a lower extent, from RBOHH and RBOHJ activities that produce superoxide ions [42,55,56] that are further converted chemically or enzymatically to H_2_O_2_. Then, part of H_2_O_2_ might be transported from the apoplast to the cytoplasm side by specific PIPs, as it was shown in other plant cell types (e.g., in stomata and epidermal cells), in response to diverse stimuli [45,46,47]. When apoplastic PRX protein levels and, therefore, peroxidase activity are low, as in the triple mutant *prx01,44,73*, high levels of H_2_O_2_ accumulate in the apoplast, triggering cell wall loosening through the oxidative cycle (OC). This affects growth homeostasis and inhibits expansion by decreasing root hair growth and cell wall thickness (Appendix A). Concomitantly, deficient PRX activity in the apoplast also triggers lower H_2_O_2_ levels in the cytoplasm of growing root hairs. This is in agreement with the fact that exogenously supplied H_2_O_2_ inhibited root hair polar expansion, whereas treatment with ROS scavengers (e.g., ascorbic acid) caused root hair bursting [57], reinforcing the notion that _apo_ROS modulates cell growth by impacting cell wall properties. Our results suggest that either low or high levels of apoplastic class-III PRXs in the root hair cell walls affect ROS homeostasis and cell wall assembly, with a detrimental effect on cell expansion. Changes in ROS homeostasis produced by altered levels of these apoplastic PRXs might affect the secretion, targeting, and possibly the crosslinking of cell wall components, including EXTs, affecting root hair elongation [40].

### 3.2. Unique Functions of the EXT-PRXs in the Root Hair Cell Walls

Currently, most of the 73 apoplastic class-III PRXs in *Arabidopsis thaliana* have no assigned biological function. In this work, we have characterized three related EXT-PRXs, PRX01, PRX44, and PRX73, that function in ROS homeostasis and potentially in EXT assembly during root hair growth. These PRXs might control Tyr crosslinking in EXTs and related *O*-glycoproteins and modify its secretion and assembly in the nascent tip cell walls. Using modeling and docking approaches, we were able to measure the in silico interactions of these PRXs with single-chain EXT substrates. The *O*-glycosylation levels in the EXT peptides clearly modified the interactions with these PRXs. PROLYL 4-HYDROXYLASE (P4H5), PEPTIDYL-SER GALACTOSYLTRANSFERASE (SGT1/SERGT1), and REDUCE RESIDUAL ARABINOSE 3 (RRA3) are key enzymes that modify EXT hydroxylation (P_4_H_5_) and EXT *O*-glycosylation (SERGT1 and RRA3) [2]. Specifically, it was shown that P_4_H_5_ is a 2-oxoglutarate (2OG) dioxygenase that catalyzes the formation of trans-4-hydroxyproline (Hyp/O) from peptidyl-proline preferentially in an EXT context, allowing these proteins to be *O*-glycosylated [16,18]. In the case of RRA3, together with RRA1–RRA2 homologous proteins [16,58], they are thought to transfer the second arabinose to the glycan chain (composed by 4–5 units of l-arabinofuranose) attached to the Hyp in the EXT peptides. SERGT1 add the single galactose units to the serine in the repetitive motif of Ser-(Pro)_3–5_ present in EXT and EXT-related proteins (Saito et al. 2014). All these lines of evidence indicate that PRX01, PRX44, and PRX73 are important enzymes that could be involved in the *O*-glycosylated EXT assembly during root hair growth. Similarly, we recently characterized PRX62 and PRX69 as key apoplastic PRXs that modulate ROS homeostasis and cell wall EXT insolubilization linked to root hair elongation at low temperature [40]. Collectively, these results support a predominant role of ROS homeostasis partially regulated by specific PRXs as a key regulator of polar root hair elongation. From an evolutionary perspective, all the putative EXT-PRXs (previously identified based on in vitro evidence or immunolabeling) do not cluster together in the phylogenetic tree of class-III PRXs, suggesting that plant-related EXT-PRXs might have evolved several times in parallel during Tracheophyte evolution. Interestingly, as a convergent evolutionary extracellular assembly, hydroxyproline-rich collagen class-IV, similar to the green EXT linage and related glycoproteins, is also crosslinked by the activity of a specific class of animal heme peroxidases (named peroxidasin or PXDN) to form insoluble extracellular networks [59,60]. While the biophysical properties of collagen IV allow the correct development and function of multicellular tissues in all animal phyla [61], EXT assemblies also have key functions in several plant cell expansion and morphogenesis processes [1,2,3,4,5,7,12,18,19,20,51]. This might imply that crosslinked extracellular matrices based on hydroxyproline-rich polymers (e.g., collagens and EXTs) have evolved more than once during eukaryotic evolution, providing mechanical support to single and multiple cellular tissues. Further analyses are required to establish how these described EXT-PRXs catalyze Tyr crosslinks on EXTs at the molecular level and how this assembly process is regulated during polar cell expansion.

## 4. Materials and Methods

### 4.1. Plant and Growth Conditions

*Arabidopsis thaliana* Columbia-0 (Col-0) was used as the wild-class (Wt) genotype in all experiments. All mutants and transgenic lines tested are in this genetic background. Seedlings were germinated on agar plates in a Percival incubator at 22 °C in a growth room for 7 days at 140 μmol m^−2^ s^−1^ light intensity. Plants were transferred to soil for growth under the same conditions. For identification of T-DNA knockout lines, genomic DNA was extracted from rosette leaves. Confirmation by PCR of a single and multiple T-DNA insertions in the target PRX genes was performed using an insertion-specific LBb1.3 primer in addition to one gene-specific primer. To ensure gene disruptions, PCR was also run using two gene-specific primers, expecting bands corresponding to fragments larger than in WT. We isolated homozygous lines for PRX01 (AT1G05240, *prx01-2*, and Salk_103597), PRX44 (AT4G26010, *prx44-2*, and Salk_057222), and PRX73 (AT5G67400, *prx73-3*, and Salk_009296). SERGT1 (*sergt1-1* SALK_054682), *rra3* (GABI_233B05) [16], and *p4h5* T-DNA mutant [16] were isolated and described previously. Double and triple mutants were generated by manual crosses of the corresponding single mutants [7]. All the mutant lines used in this study are described in Appendix A.

### 4.2. PRX:GFP and 35S:PRX-GFP Lines

Vectors based on the Gateway cloning technology (Invitrogen) were used for all manipulations. Constitutive expression of PRXs-GFP-tagged lines were achieved in plant destination vector pMDC83. cDNA PRXs sequences were PCR-amplified with AttB recombination sites. PCR products were then recombined first in pDONR207 and transferred into pGWB83. To generate transcriptional reporter, the PRX promoter regions (2 Kb) were amplified and recombined first in pDONR207 and transferred into pMDC111. All the transgenic lines used in this study are described in Appendix A.

### 4.3. SS-TOM and SS-TOM-EXT-LONG Constructs

The binary vector pART27, encoding tdTomato (TOM), secreted with the secretory signal sequence from tomato polygalacturonase, and expressed by the constitutive CaMV 35S promoter (pART-SS-TOM), was the kind gift of Dr. Jocelyn Rose, Cornell University. The entire reporter protein construct was excised from pART-SS-TOM by digesting with NotI. The resulting fragments were gel-purified with the QIAquick Gel Extraction Kit and ligated using T4 DNA Ligase (New England Biolabs) into dephosphorylated pBlueScript KS+ that had also been digested with NotI and gel-purified to make pBS-SS-TOM. The plasmid was confirmed by sequencing with primers 35S-FP (5′-CCTTCGCAAGACCCTTCCTC-3′) and OCS-RP (5′-CGTGCACAACAGAATTGAAAGC-3′). The sequence of the EXT domain from SlPEX1 (NCBI accession AF159296) was synthesized and cloned by GenScript into pUC57 (pUC57-EXT). The plasmid pBS-SS-TOM-EXT-LONG was made by digesting pUC57-EXT and pBS-SS-TOM with NdeI and SgrAI, followed by gel purification of the 2243 bp band from pUC57-EXT and the 5545 bp band from pBS-SS-TOM, and ligation of the two gel-purified fragments. The pBS-SS-TOM-EXT-LONG plasmid was confirmed by sequencing with 35S-FP, OCS-RP, and tdt-seq-FP (5′-CCCGTTCAATTGCCTGGT-3′). Both pBS plasmids were also confirmed by digestion. The binary vector pART-SS-TOM-EXT-LONG was made by gel purifying the NotI insert fragment from the pBS-SS-TOM-Long EXT plasmid and ligating it with pART-SS-TOM backbone that had been digested with NotI, gel purified, and dephosphorylated. This plasmid was confirmed by sequencing. The construct SS-TOM and SS-TOM-Long-EXT were transformed into Arabidopsis plants. The secretory sequence (SS) from tomato polygalacturonase is MVIQRNSILLLIIIFASSISTCRSGT (2.8 kDa) and the EXT-Long domain sequence with six alanine cluster is BAAAAAAACTLPSLKNFTFSKNIFESMDETCRPSESKQVKIDGNENCLGGRSEQRTEKECFPVVSKPVDCSKGHCGVSREGQSPKDPPKTVTPPKPSTPTTPKPNPSPPPPKTLPPPPKTSPPPPVHSPPPPPVASPPPPVHSPPPPVASPPPPVHSPPPPPVASPPPPVHSPPPPVASPPPPVHSPPPPVHSPPPPVASPPPPVHSPPPPVHSPPPPVHSPPPPVHSPPPPVHSPPPPVASPPPPVHSPPPPVHSPPPPVHSPPPPVASPPPPVHSPPPPPPVASPPPPVHSPPPPVASPPPPVHSPPPPVASPPPPVHSPPPPVHSPPPPVHSPPPPVASPPPALVFSPPPPVHSPPPPAPVMSPPPPTFEDALPPTLGSLYASPPPPIFQGY × 395-(39.9 kDa). The predicted molecular size for SS-TOM protein is 54.2 kDa and for SS-TOM-EXT-Long Mw is 97.4 kDa. All the transgenic lines used in this study are described in Appendix A.

### 4.4. Root Hair Phenotype

For quantitative analysis of root hair phenotypes in *prx01,44,73* mutant, 35S:PRX-GFP lines, and Wt Col-0, 200 fully elongated root hairs were measured (n roots = 20–30) from seedlings grown on vertical plates for 10 days. Values are reported as the mean ±SD using the Image J software. Measurements were made after 7 days. Images were captured with an Olympus SZX7 Zoom microscope equipped with a Q-Colors digital camera.

### 4.5. Confocal Imaging

Root hairs were ratio imaged with the Zeiss LSM 710 laser scanning confocal microscope (Carl Zeiss, Jena, Germany) using a 40× oil-immersion, 1.2 numerical aperture. EGFP (473–505 nm) emission was collected using a 458-nm primary dichroic mirror and the meta-detector of the microscope. Bright-field images were acquired simultaneously using the transmission detector of the microscope. Fluorescence intensity was measured in 7 µm region of interest (ROI) at the root hair apex. For the lines SS-TOMATO and SS-TOMATO-EXT-LONG in the different genetic backgrounds, roots were plasmolyzed with an 8% mannitol solution and the scanning was performed using Zeiss LSM 510 META (Zeiss, Germany) (excitation: 543 nm argon laser; emission: 560–600 nm, Zeiss Plain Apochromat 63×/1.4 -Oil objective). The GFP signal and TOM cell wall signal at the RH tip were quantified using the ImageJ software. Fluorescence AU were expressed as the mean ± SD using the GraphPad Prism 8.0.1 (San Diego, CA, USA) statistical analysis software. Results are representative of two independent experiments, each involving 10 roots, and approximately between 10 and 20 hairs per root were observed.

### 4.6. Peroxidase Activity

Soluble proteins were extracted from roots grown on agar plates in a Percival incubator at 22 °C in a growth room for 7 days at 140 μmol m^−2^ s^−1^ light intensity by grinding in 20 mM HEPES, pH 7.0, containing 1 mM EGTA, 10 mM ascorbic acid, and PVP PolyclarAT (100 mg g^−1^ fresh material; Sigma, Buchs, Switzerland). The extract was centrifuged twice for 10 min at 10,000 g. Each extract was assayed for protein levels with the Bio-Rad assay (Bio-Rad Hercules, Hercules, CA, United States). PRX activity was measured at 25 °C by following the oxidation of 8 mM guaiacol (Fluka, Charlotte, North Carolina, United States) at 470 nm in the presence of 2 mM H_2_O_2_ (Carlo Erba Val-de-Reuil, Val-de-Reuil, France) in a phosphate buffer (200 mM, pH6.0). Values are the mean of three replicates ± SD.

### 4.7. Cytoplasmic ROS (cytROS) Measurements

2′,7′-dichlorodihydrofluorescein diacetate (H_2_DCF-DA) is used as a cell-permeable fluorogenic probe to quantify reactive oxygen species (ROS). H_2_DCFDA diffuses into cells and is deacetylated by cellular esterases to form 2′,7′-dichlorodihydrofluorescein (H_2_DCF). In the presence of ROS, predominantly H_2_O_2_, H_2_DCF is rapidly oxidized to 2′,7′-dichlorofluorescein (DCF), which is highly fluorescent, with excitation and emission wavelengths of 498 and 522 nm, respectively. To measure cytoplasmic ROS in root hair cells, growth of Arabidopsis seeds on a plate was conducted with 1% sterile agar for 7 days in a chamber at 22 °C with continuous light. These seedlings were incubated in darkness on a slide for 10 min with 50 μM H_2_DCFDA at room temperature. Samples were observed with Zeiss Imager A2 Epifluorescence. A 10× objective was used, 0.30 N.A., and exposure time 80–500 ms. Images were analyzed using ImageJ 1.50 b software. To measure ROS mean, a circular region of interest (ROI) (r = 2.5) was chosen in the tip zone of the root hair. All root hairs of six seedlings per genotype were analyzed. The reported values are the mean ± standard deviation (mean ± SD).

### 4.8. Apoplastic ROS (apoROS) Measurements

To measure apoplastic ROS in root hair cells, roots of 7-day-old seedlings were incubated with 50 µM Amplex™ UltraRed Reagent (AUR, Molecular Probes Eugene, Eugene, Oregon, United States) for 20 min in dark conditions and rinsed with liquid MS. Root hairs were imaged with a Zeiss LSM5 Pascal laser scanning confocal microscope. The fluorescence emission of oxidized AUR in the apoplast of root hair cells was observed between 585 and 610 nm using 543 nm argon laser excitation, 40× objective, N/A = 1.2. The intensity of fluorescence was quantified on digital images using ImageJ software. Quantification of the AUR probing fluorescence signal was restricted to apoplastic spaces at the root hair tip. The measurements were performed in three independent experiments (n ≥ 5) with the same microscopic settings.

### 4.9. Phylogenetic Analysis

A total of 73 class-III PRX protein sequences from A. thaliana, 2 putative lignin class-III PRXs from *Zinnia elegans*, and 4 putative extensin class-III PRXs from *Lupinus album*, *Lycopersicum esculentum*, *Phaseolus vulgaris*, and *Vitis vinifera* have been aligned with ClustalW and the tree constructed using the neighbor-joining method [62]. The analyses were conducted in MEGA7 [63]. All protein sequences are available using their ID number (http://peroxibase.toulouse.inra.fr) Accessed on 1 March 2022 [64].

### 4.10. Co-Expression Analysis Network

Co-expression networks for RSL4 root hair genes were identified from PlaNet (http://aranet.mpimp-golm.mpg.de) Accessed on 1 March 2022 and trimmed to facilitate readability (Mutwill et al., 2011). Each co-expression of interest was confirmed independently using the expression angler tool from Botany Array Resource BAR (http://bar.utoronto.ca/ntools/cgi-bin/ntools_expression_angler.cgi) Accessed on 1 February 2018 and ATTED-II (http://atted.jp). Accessed on 1 February 2018 Only those genes that are connected with genes of interest are included.

### 4.11. Tyr-Crosslinking Analysis

Alcohol-insoluble residues of root tissues obtained from *prx01,44,73* mutants, Col-0, and 35S:PRX44-GFP lines were hydrolyzed in 6 N HCl (aqueous) with 10 mM phenol (2 mg mL^−1^; 110 °C; 20 h). Hydrolysates were dried under a steady stream of nitrogen (gas) and then redissolved at 10 mg ml-1 in water. The hydrolysates were fractionated by gel permeation chromatography on a polyhydroxyethyl A column (inner diameter, 9.4 × 200 mm, 10 nm pore size, Poly LC Inc., Columbia, MD, USA) equilibrated in 50 mM formic acid and eluted isocratically at a flow rate of 0.8 mL min^−1^. UV absorbance was monitored at 280 nm. The amounts of Tyr and IDT in the hydrolysates were then determined by comparison with peak areas of authentic Tyr and IDT standards. Response factors were determined from three level calibrations with the Tyr and IDT standards.

### 4.12. Immunoblot Analysis

Plant material (100 mg of root from 15-day-old seedlings grown as indicated before) was collected in a microfuge tube and ground in liquid nitrogen with 400 μL of protein extraction buffer (125 mM Tris-Cl, pH. 4.4, 2% (*w*/*v*) SDS, 10% (*v*/*v*) glycerol, 6 M UREA, 1% (*v*/*v*) β-mercaptoethanol, 1 mM PMSF). Samples were immediately transferred to ice. After centrifugations at 13000 rpm at 4 °C for 20 min, supernatant was moved to a new 1.5 mL tube and equal volumes of Laemmli buffer (125 mM Tris-Cl, pH. 7.4, 4% (*w*/*v*) SDS, 40% (*v*/*v*) glycerol, 10% (*v*/*v*) β-mercaptoethanol, 0.002% (*w*/v) bromophenol blue) were added. The samples (0.5–1.0 mg/mL of protein) were boiled for 5 min and 30 μL were loaded on 10% SDS-PAGE. The proteins were separated by electrophoresis and transferred to nitrocellulose membranes. Anti-GFP mouse IgG (clones 7.1 and 13.1; Roche Applied Science Penzberg, Germany) was used at a dilution of 1:2000 and it was visualized by incubation with goat anti-mouse IgG secondary antibodies conjugated to horseradish peroxidase (1:2000), followed by a chemiluminescence reaction (Clarity Western ECL Substrate; Bio-rad Hercules CA, United States). For the SS-TOM lines analysis, proteins were extracted in 2× SDS buffer (4% SDS, 125 mM Tris pH 6.8, 20% glycerol, 0.01% bromophenol blue, 50 mM dithiothreitol (DTT)), using 10 μL of buffer per mg of plant tissues of Wt Col-0, transgenic lines 35S:SS-TOM, and 35S:SS-TOM-EXT-LONG. Two transgenic lines were analyzed. A total of 10 μL of supernatant of each protein extract was run into a 12% polyacrylamide gel during one hour at 200 V, and then transferred to a PVDF membrane. PVDF was blocked with 5% milk in TBST (Tris-HCl 10 mM, pH 7,4, NaCl 150 mM, Tween-20 at 0.05%) for 1 h at 4 °C, and then washed four times during 15 min in TBST. An anti-RFP (A00682, GenScript New Jersey, United States) was used as the primary antibody (that recognizes TOM protein) overnight at 4 °C. After four washes of 15 min each in TBST at room temperature, it was incubated two hours with a secondary antibody anti-rabbit (goat) conjugated with alkaline phosphatase (A3687, Sigma) in a 1:2500 dilution with TBST. This was followed by four washes of 15 min each in TBST at room temperature. Finally, 10 μL of alkaline phosphatase (100 mM Tris-HCl pH 9.5, 100 mM NaCl, 3 mM MgCl_2_) containing 80 μL NBT (Sigma, Saint Louis, MO, United States) (35 mg/mL in 70% DMSO) and 30 μL de BCIP (Sigma) (50 mg/mL in 100% de DMSO) was used.

### 4.13. Transmission Electron Microscopy of Root Hair Cell Walls

Seeds were germinated on 0.2× MS, 1% sucrose, 0.8% agar. Seven days after germination, seedlings were transferred to new 0.2× MS, 1% sucrose, 0.8% agar plates. After 4 additional days, 1-mm root segments with root hairs were fixed in 2% glutaraldehyde in 0.1 M cacodylate buffer pH 7.4. Samples were rinsed in cacodylate buffer and post-fixed in 2% OsO_4_. After dehydration in ethanol and acetone, samples were infiltrated in Epon resin (Ted Pella, Redding, CA, USA). Polymerization was performed at 60 °C. Sections were stained with 2% uranyl acetate in 70% methanol, followed by Reynold’s lead citrate (2.6% lead nitrate and 3.5% sodium citrate (pH 12.0)) and observed in a Tecnai 12 electron microscope. Quantitative analysis of cell wall thickness was performed using FIJI.

### 4.14. Modeling and Molecular Docking between PRXs and EXTs

Modeling and molecular docking: cDNA sequences of PRXs were retrieved from TAIR (PRX01: AT1G05240, PRX44: AT4G26010, PRX62: AT5G39580, PRX69: AT5G64100, PRX73: AT5G67400) and NCBI Nucleotide DB (PRX24Gv:*Vitis vinifera* peroxidase 24, GvEP1, LOC100254434). Homology modeling was performed for all PRXs using modeller 9.14 [65], using the crystal structures 1PA2, 3HDL, 1QO4, and 1HCH as templates, available at the protein data bank. In total, 100 structures were generated for each protein and the best scoring one (according to DOPE score) was chosen. The receptor for the docking runs was generated by the prepare_receptor4 script from autodock suite, adding hydrogens and constructing bonds. Peptides based on the sequence PYYSPSPKVYYPPPSSYVYPPPPS were used, replacing proline by hydroxyproline, and/or adding O-Hyp glycosylation with up to four arabinoses per hydroxyproline in the fully glycosylated peptide and a galactose on the serine, as it is usual in plant O-Hyp https://www.ncbi.nlm.nih.gov/pmc/articles/PMC5045529/ Accessed on 1 February 2020. Ligand starting structure was generated as the most stable structure by molecular dynamics [7]. All ligand bonds were set to be able to rotate. Docking was performed in two steps, using Autodock vina [65]. First, an exploratory search over the whole protein surface (exhaustiveness 4) was conducted, followed by a more exhaustive one (exhaustiveness 8), reducing the search space to a 75 × 75 × 75 box centered over the most frequent binding site found in the former run.

### 4.15. EXT Conformational Coarse-Grained Model

The use of coarse-grained (CG) molecular dynamics (MD) allowed collection of long timescale trajectories. System reduction is significant when compared to all atom models, approximately reducing on order of magnitude in particle number. In addition, a longer integration time step can be used. Protein residues and coarse-grained solvent parameters correspond to the SIRAH model [66], while ad hoc specific glycan parameters were developed. The CG force field parameters developed correspond to arabinofuranose and galactopyranose (Appendix A). Triple-helix systems were simulated, both in the non-glycosylated and fully *O*-glycosylated states, where all the hydroxyprolines are bound to a tetrasaccharide of arabinofuranoses, and specific serine residues contain one galactopyranose molecule. They were immersed in WT4 GC solvent box that was constructed to be 2 nm apart from the extensin fiber, and periodic boundary conditions were employed. Coarse-grained ions were also included to achieve electroneutrality and 0.15 M ionic strength. All simulations were performed using the GROMACS MD package at constant temperature and pressure, using the Berendsen thermostat (respectively) and Parrinello–Rahman barostat [67], and a 10 fs time step. The obtained trajectories were analyzed using the Mdtraj python package [68] and visualized with Visual Molecular Dynamics (VMD) 1.9.1 [69]. Volume measurements were performed using a Convex Hull algorithm implemented in NumPy [70], and average diameter calculations were derived from this quantity using simple geometric arguments.

## Figures and Tables

**Figure 1 ijms-23-05375-f001:**
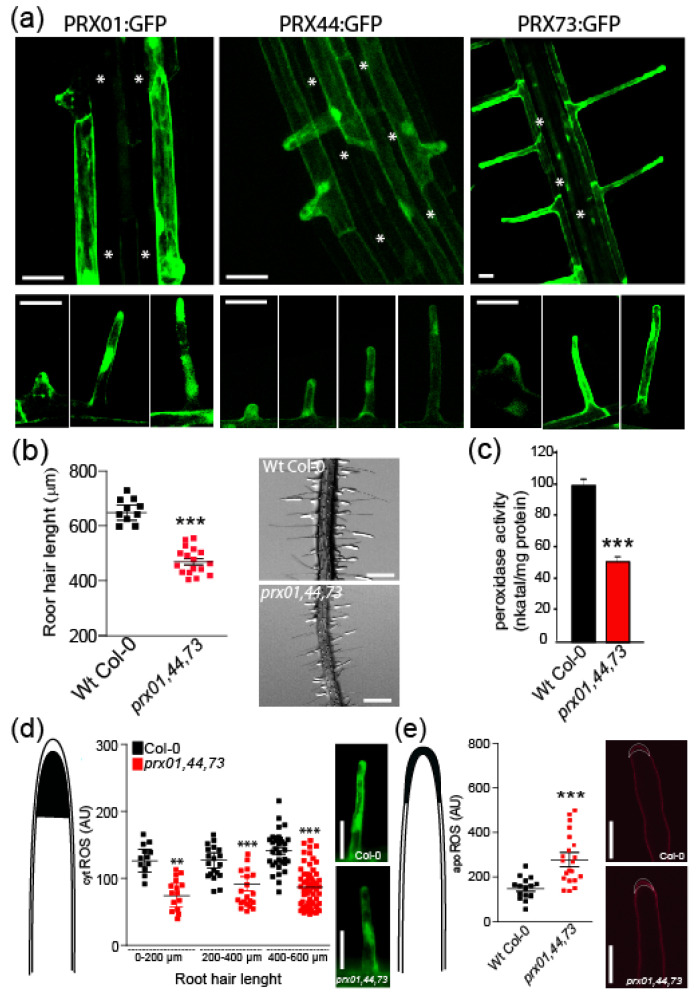
Characterization of root-hair-specific PRX01, PRX44, and PRX73 expression and mutant analysis. (**a**) GFP-tagged transcriptional reporters of PRX01, PRX44, and PRX73 show expression in the root elongation zone and specifically in root hairs (bottom). Scale bar = 200 μm. (*) indicates atrichoblast cell files, which lack GFP expression. (**b**) Root hair length phenotype of Wt Col-0 and the *prx01,44,73* triple mutant. Left, box-plot of root hair length. Horizontal lines show the means. P-value determined by one-way ANOVA, (***) *p* < 0.001. Right, bright-field images exemplifying the root hair phenotype in each genotype. Scale bars = 500 μm. (**c**) Peroxidase activity in Wt Col-0 and *prx01,44,73* triple mutant roots. Enzyme activity values (expressed as nkatal/mg protein) are shown as the mean of three replicates ± SD. P-value determined by one-way ANOVA, (***) *p* < 0.001. (**d**) Cytoplasmic ROS levels measured with H2DCF-DA in Wt and *prx01,44,73* triple mutant root hairs. Right, horizontal lines show the means. *p*-values determined by two-way ANOVA, (***) *p* < 0.001 and (**) *p* < 0.01. Right, fluorescence images exemplifying _cyt_ROS detection in root hairs. Scale bars, 20 μm. (**e**) Apoplastic ROS levels measured with Amplex™ UltraRed (AUR) in Wt and *prx01,44,73* triple mutant root hairs. ROS signal was quantified from the root hair cell tip. Left, box-plot of _apo_ROS values. Horizontal lines show the means. *p*-value determined by one-way ANOVA, (***) *p* < 0.001. Right, fluorescence images exemplifying apoROS detection in root hair apoplast. Scale bars, 200 μm.

**Figure 2 ijms-23-05375-f002:**
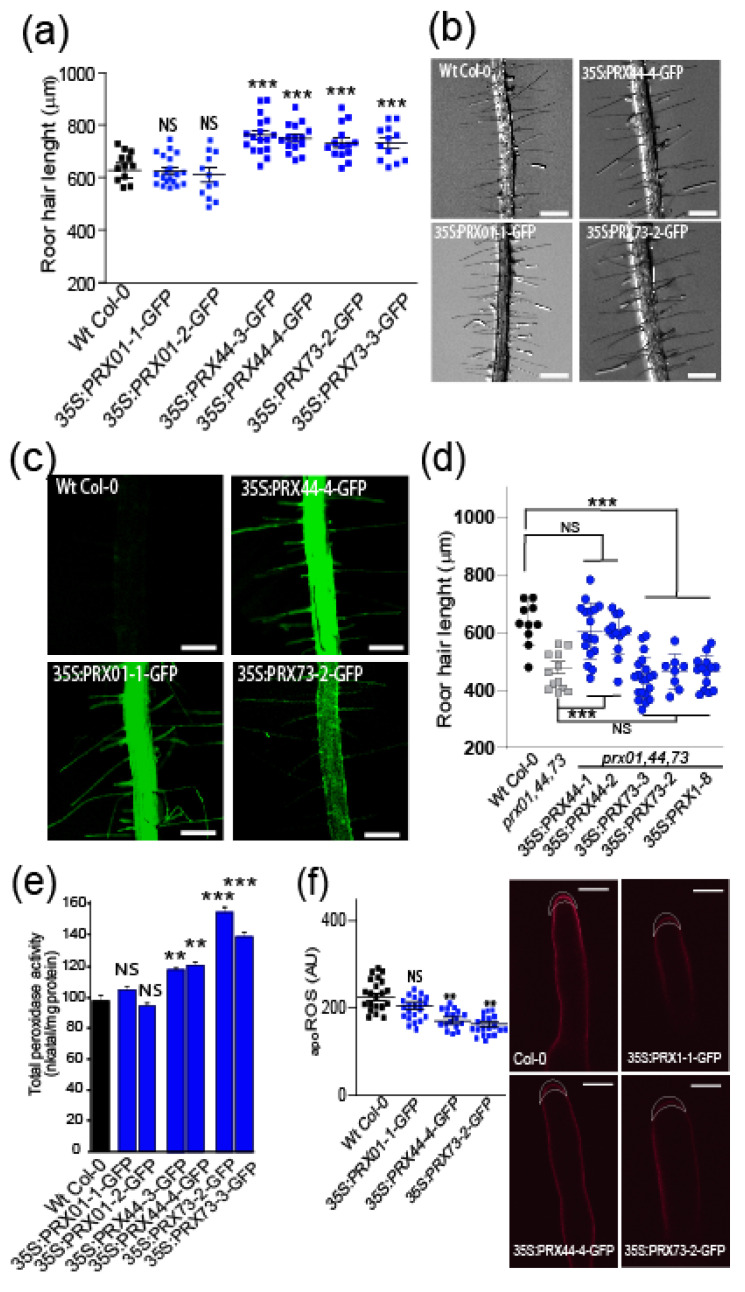
Partially redundant functions of PRX01, PRX44, and PRX73 in promoting root hair growth linked to their peroxidase activity. (**a**) Root hair length phenotype of Wt Col-0 and 35S:PRX lines (in Wt Col-0 background). Box-plot of root hair length. Horizontal lines show the means. *p*-values determined by one-way ANOVA, (***) *p* < 0.001, (NS) not significantly different. (**b**) Bright-field images exemplifying the root hair phenotype analyzed in (**a**). Scale bar = 0.5 mm. (**c**) Expression of GFP-tagged 35S:PRX01, 35S:PRX44, and 35S:PRX73 in root hair cells. (**d**) Root hair length phenotype of Wt Col-0, *prx01,44,73* triple mutant, and 35S:PRX lines (in *prx01,44,73* triple mutant background). Left, box-plot of root hair length. Horizontal lines show the means. *p*-value determined by one-way ANOVA, (***) *p* < 0.001. (**e**) Assays of total peroxidase activity in Wt and 35S:PRX lines (in Wt Col-0 background). Enzyme activity (expressed in nkatal/mg protein) was determined by a guaiacol oxidation-based assay. Values are the mean of three replicates ± SD. *p*-values determined by one-way ANOVA, (***) *p* < 0.001, (**) *p* < 0.01, (NS) not significantly different. (**f**) Apoplastic ROS levels measured with Amplex™ UltraRed (AUR) in Wt Col-0 and 35S:PRX lines (in Wt Col-0 background). ROS signal was quantified from the root hair cell tip. Left, box-plot of _apo_ROS values. Horizontal lines show the means. *p*-values determined by one-way ANOVA, (**) *p* < 0.01, (NS) not significantly different. Right, fluorescence images exemplifying _apo_ROS detection in root hair apoplast. Scale bar = 10 μm.

**Figure 3 ijms-23-05375-f003:**
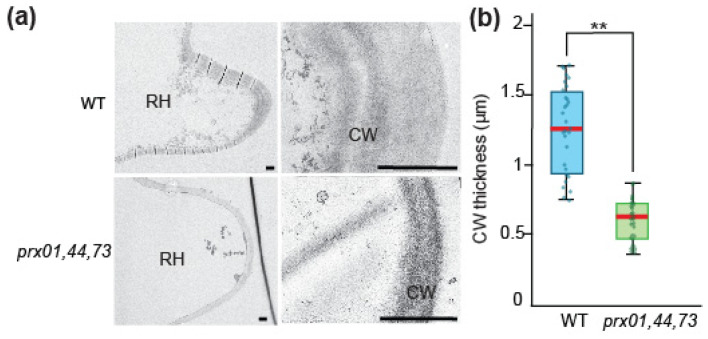
Effect of PRXs on cell wall thickness in root hair tips. (**a**) Transmission electron micrographs of developing root hair tips from Wt Col-0 and *prx01,44,73* triple mutant. Representative overviews of young root hairs (RH) of each genotype and a detail of the cell wall at the root hair tip (CW) are shown. Scale bar = 1 mm. (**b**) Box and whisker plot showing variation in cell wall thickness measured at the root hair tip of the three genotypes. Graph shows quartiles, the middle line indicates the median and whiskers show the upper and lower fences. Dots indicate individual data points. (**) *p* ˂ 0.001 determined by *t*-test.

**Figure 4 ijms-23-05375-f004:**
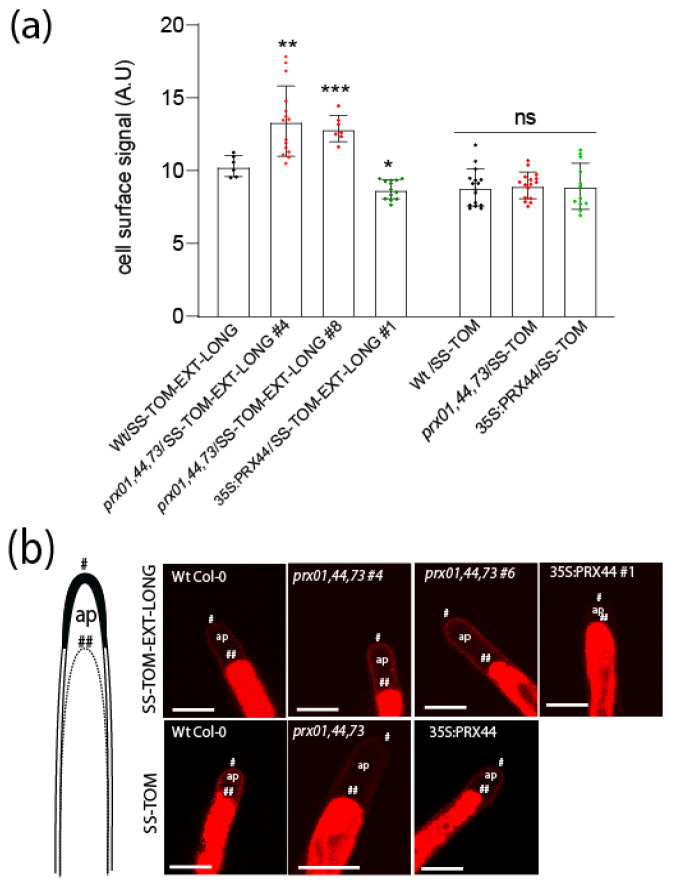
EXT reporter stabilization in root hair apical cell wall depends on PRX01, PRX44, and PRX73. (**a**) Signal of SS-TOM and SS-TOM-EXT-LONG in the apical zone of root hairs in Wt (Col-0), *prx01,44,73* triple mutant, and 35S:PRX44 lines. Each point is the signal derived from a single root hair tip. Fluorescence AU data are the mean ± SD (N = 10 root hairs), two-way ANOVA, followed by a Tukey–Kramer test; (*) *p* < 0.05, (**) *p* < 0.01, (***) *p* < 0.001. Results are representative of two independent experiments. Asterisks on the graph indicate significant differences between genotypes. NS = non-significant differences. (**b**) In the images: (*) indicates cell surface, including the plant cell walls, (**) indicates the retraction of the plasma membrane, (ap) apoplastic space delimitated between the plant cell wall and the retracted plasma membrane. Scale bars = 15 μm.

**Table 1 ijms-23-05375-t001:** Peptidyl-Tyr and iso-dityrosine (IDT) contents in cell walls isolated from Wt Col-0, *prx01,44,73* triple mutant, 35S:PRXs lines, and mutant lines with under-glycosylated EXTs. *p*-values were determined by one-way ANOVA, (***) *p* ˂ 0.001, (**) *p* ˂ 0.01. STD = standard deviation. Values significantly different than Wt Col-0 are highlighted in blue if higher and in light blue if lower than Wt Col-0.

	ng Tyr/μg CW (STD)	ng IDT/μg CW (STD)
Wt Col-0	7.799 ± 0.26	0.853 ± 0.08
*prx01,44,73*	9.588 ± 0.31 **	0.963 ± 0.02
*35S:PRX44*	8.649 ± 0.07	0.953 ± 0.04
*35S:PRX73*	8.700 ± 0.12	1.042 ± 0.02 **
*under O-glycosylated EXTs*		
*sergt1-1 rra3*	3.530 ± 0.08 ***	0.235 ± 0.01 ***
*p4h5 sergt1-1*	3.766 ± 0.06 ***	0.225 ± 0.02 ***

## Data Availability

The data presented in this study are available on request from the corresponding author.

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
