# Peer review of "Class III Peroxidases PRX01, PRX44, and PRX73 Control Root Hair Growth in Arabidopsis thaliana"

_ijms, 2022, doi:10.3390/ijms23105375_

Round 1
Reviewer 1 Report
Dear Authors,
I have an opportunity to review paper entitled “Class III peroxidases PRX01, PRX44, and PRX73 controls root hair growth in Arabidopsis thaliana” submitted to IJMS MDPI.
Authors concentrated on peroxidases (PRXs) role in polar expansions root hair growth condition. Moreover, Authors identified and characterized three root hair-specific putative EXT-PRXs, PRX01, PRX44, and PRX73.
- In my opinion introduction provided sufficient background;
- Obtained results have potential and are interesting for plant biologist
- Presenting results are in general clearly described and well designed
- Please, underline clearly the aim of the study and /or potential hypothesis, because in current form the reader found right away the main brief-results;
- Generally, I would like to underline a very good quality of microphotographic documentation, but still some improvements in figure quality are needed, some examples:
-Figure 1- panel D and E – if in E section we can find fluorescent microphotos, it should be also added in panel D;
-Figure 3 – in my opinion the it is a figure with the worse electron microscopy microphotographs- I suggest to change it to enlarged and with better quality;-it is a very important result in the context of influence on cell wall structure;
Some figure caption are like’ cut-off’ in pdf version of manuscript – like in figure 3 or – please, correct it;
-On the other side-I did not find in manuscript figure 4 caption- Is it possible? -and this figure is unnaturally ‘elongated’;
- Materials and methods chapter are clearly and adequately described;
- Since Authors have propose some kind of ‘working model’ , I suggest to compose some graphical diagrams with relations- maybe it suggest also future prospect coming from obtained results;
- Unfortunately, the discussion chapter is insufficient in current form; I have a question: do they exist some findings about the role class III peroxidases in plant- rhizosphere’s microbe interactions?
Minor comments: I suggest to unified fonts and font size in the whole manuscript;
Moreover, the references list as well as citations in the text should be arrange according IJMS rules- Please, correct it;
Reviewer 2 Report
This is a well written Manuscript. All the conclusions are supported by the obtained results. In my opinion, this Manuscript can be published as it is without any corrections.
Reviewer 3 Report
The manuscript by Marzol et al. presented the role of three Class III apoplastic PRXs, PRX01, PRX44 and PRX73, as key enzymes that most likely play their roles in the insolubilization of cell wall EXTs in growing root hair cells.
I like it and recommend it for publication after several minor points are addressed:
1) To improve readability of the Results section, I strongly recommend putting several subheadings to the text to divide it into several sections.
2) Figures 1DE, and Figure 2F (these parts in particular) are not referred to in the main text. Please check and refer to them.
3) The size of the scale bar in Figure 1E is not given, please check and add it.
4) In Figure 2C, the fluorescence should be quantified by a software and the acquired numbers should be statistically compared.
5) Figure 4 has a very short legend. The legend has to be improved to become more explanatory also with respect to image parts.
